# Biclustering reveals potential knee OA phenotypes in exploratory analyses: Data from the Osteoarthritis Initiative

**Amanda E. Nelson**[1]*, **Thomas H. Keefe**[2], **Todd A. Schwartz**[1,3], **Leigh F. Callahan**[1], **Richard F. Loeser**[1], **Yvonne M. Golightly**[1,4], **Liubov Arbeeva**[1], **J. S. Marron**[2]

**1** Thurston Arthritis Research Center, University of North Carolina at Chapel Hill, Chapel Hill, North Carolina, United States of America, **2** Statistics and Operations Research, University of North Carolina at Chapel Hill, Chapel Hill, North Carolina, United States of America, **3** Department of Biostatistics, Gillings School of Global Public Health, University of North Carolina at Chapel Hill, Chapel Hill, North Carolina, United States of America, **4** Department of Epidemiology, Gillings School of Global Public Health, University of North Carolina at Chapel Hill, Chapel Hill, North Carolina, United States of America

* aenelson@med.unc.edu

## Abstract

### Objective

To apply biclustering, a methodology originally developed for analysis of gene expression data, to simultaneously cluster observations and clinical features to explore candidate phenotypes of knee osteoarthritis (KOA) for the first time.

### Methods

Data from the baseline Osteoarthritis Initiative (OAI) visit were cleaned, transformed, and standardized as indicated (leaving 6461 knees with 86 features). Biclustering produced submatrices of the overall data matrix, representing similar observations across a subset of variables. Statistical validation was determined using the novel SigClust procedure. After identifying biclusters, relationships with key outcome measures were assessed, including progression of radiographic KOA, total knee arthroplasty, loss of joint space width, and worsening Western Ontario and McMaster Universities Osteoarthritis Index (WOMAC) scores, over 96 months of follow-up.

### Results

The final analytic set included 6461 knees from 3330 individuals (mean age 61 years, mean body mass index 28 kg/m², 57% women and 86% White). We identified 6 mutually exclusive biclusters characterized by different feature profiles at baseline, particularly related to symptoms and function. Biclusters represented overall better (#1), similar (#2, 3, 6), and poorer (#4, 5) prognosis compared to the overall cohort of knees, respectively. In general, knees in biclusters #4 and 5 had more structural progression (based on Kellgren-Lawrence grade, total knee arthroplasty, and loss of joint space width) but tended to have an improvement in WOMAC pain scores over time. In contrast, knees in bicluster #1 had less incident and

**Data Availability Statement:** All relevant data for this analysis are available at https://nda.nih.gov/study.html?id=911, DOI: 10.15154/1519056. All

OAI data are publicly available at: https://data-archive.nimh.nih.gov/oai/.

**Funding:** AEN, THK, JSM: NIH/NIAMS R21AR074685 AEN, LFC, TAS, RFL: NIH/NIAMS P30AR072580 JSM: NSF/DMS-2113404 NIAMS: https://www.niams.nih.gov/ NSF/DMS: https://nsf.gov/ The OAI is a public–private partnership comprising 5 contracts (N01-AR-2-2258; N01-AR-2-2259; N01-AR-2-2260; N01-AR-2-2261; N01-AR-2-2262) funded by the NIH, a branch of the Department of Health and Human Services and conducted by the OAI Study Investigators. Funding partners include Merck Research Laboratories, Novartis Pharmaceuticals Corp., GlaxoSmithKline, and Pfizer Inc. The funders had no role in study design, data collection and analysis, decision to publish, or preparation of the manuscript.

**Competing interests:** The OAI is a public–private partnership comprising 5 contracts (N01-AR-2-2258; N01-AR-2-2259; N01-AR-2-2260; N01-AR-2-2261; N01-AR-2-2262) funded by the NIH, a branch of the Department of Health and Human Services and conducted by the OAI Study Investigators. Funding partners include Merck Research Laboratories, Novartis Pharmaceuticals Corp., GlaxoSmithKline, and Pfizer Inc. There are no patents, products in development or marketed products associated with this research to declare. This does not alter our adherence to PLOS ONE policies on sharing data and materials.

progressive KOA, fewer total knee arthroplasties, less loss of joint space width, and stable pain scores compared with the overall cohort.

## Significance

We identified six biclusters within the baseline OAI dataset which have varying relationships with key outcomes in KOA. Such biclusters represent potential phenotypes within the larger cohort and may suggest subgroups at greater or lesser risk of progression over time.

## Introduction

Osteoarthritis (OA) is the most common joint disease, affecting more than 1 in 5 adults in the United States, with 1 in 10 reporting limitations in function related to their arthritis [1]. Osteoarthritis (OA) is by far the most common type of arthritis, particularly knee OA (KOA) which can significantly affect mobility and is a major contributor to disability and eventual need for joint replacement [2]. In recent years, greater attention has been given to the likelihood of different phenotypes of OA, which can be considered a syndrome rather than a single disease, as it is characterized by similar signs and symptoms across patients, but these are due to differing causes and manifestations [3, 4]. The common risk factors for OA, such as aging, increased body weight, joint injury, genetics, and biomechanical factors are quite diverse. Therefore, clinical studies that do not account for this heterogeneity are inherently limited. Combining individuals with later stage features like pain and radiographic changes will mask the range of important potential pathways leading to the OA syndrome in each individual, limiting the ability of researchers to stratify risk and target treatments [5]. A range of phenotypes have been proposed, including age and cellular senescence [6], post-injury OA in younger adults [7], inflammatory predominant [8], and obesity/metabolic syndrome-associated [9], among others [10, 11]. It is apparent that pain mechanisms vary among individuals with OA [12] and therefore defining differences in pain manifestations in the various OA phenotypes [13, 14] will likely also be necessary. Identification of individuals with the greatest risk of progression could help to optimize clinical trials designed to slow or stop structural progression.

A variety of machine learning methodologies initially utilized primarily for image analysis in OA [15] are being increasingly applied to large datasets for the purpose of identifying important OA subgroups such as those mentioned above [16]. The Osteoarthritis Initiative (OAI) [17], as a publicly available rich data source, is frequently utilized in such work, which often seeks to identify patterns in magnetic resonance imaging (MRI) or radiographic images [18, 19] with or without clinical data aspects [16, 20, 21]. Novel methodologies originally developed for other large datasets, such as array or genetic data, are also increasingly being applied to large clinical datasets to explore potential phenotypes. In this paper, we used such a methodology, called biclustering [22], originally developed for analysis of gene expression data, to simultaneously cluster observations and clinical features to explore candidate phenotypes of KOA within the OAI dataset. We subsequently characterized the identified groups by important outcomes to explore relationships with structural and symptomatic progression.

## Methodology

### Data source

This research used fully anonymized and publicly available data from the OAI cohort: https://nda.nih.gov/oai with an approved data use agreement. The OAI is a multicenter, longitudinal,

prospective observational study, designed to identify risk factors for the development and progression of symptomatic KOA. The study design has been previously reported [17]. In short, 4,796 participants were recruited between 2004 and 2006 from four clinical centers and were invited to annual follow-up visits for up to 8 years. At baseline, 1,389 participants were in the progression cohort (symptomatic tibiofemoral KOA with osteophytes and frequent symptoms in one or both knees), 3,285 in the incidence cohort (no symptomatic KOA in either knee), and 122 were healthy controls (no KOA, no symptoms, and no risk factors). The starting data for these analyses, included baseline data from the OAI (n = 9592 knees), including 116 clinical and demographic features (S1 Table, https://nda.nih.gov/study.html?id=911; doi: 10.15154/1519056).

## Preparation of the baseline dataset

First, the data were downloaded, compiled, and examined in detail. We chose to treat each knee as an experimental unit, instead of each person, because it is feasible that a person's two knees could express different phenotypes of OA. People with a total knee arthroplasty (TKA) in one knee at baseline therefore only have their other non-TKA knee in the dataset. We considered skewness, kurtosis, scaling issues, outliers, and other data characteristics using marginal distribution plots [23]. To prevent the analysis from depending on measurement scale, we transformed continuous variables to have mean 0 and standard deviation 1, and because many procedures are sensitive to skewness and non-normality, we transformed them to more closely match normality. We combined transformations in a single step as previously described [24].

We removed certain variables from the dataset before analysis. In particular, we removed medication measurements for medications that are known to be unrelated to OA. We also removed variables that are redundant: for example, "WOMAC Total" is the sum of the WOMAC variables from all three subscales of pain, stiffness, and function, and "White" is redundant after including race. Three variables—"P01HRSR", "P01HRSL", and "V00HIPFX"—were excluded because they were only collected among a small subset of participants. Unordered categorical variables were encoded using an indicator variable for each level. Ordered categorical variables were encoded as integers 1, 2, 3, etc. The above processing left 86 variables.

As the final preparation step, we removed knees with any missing values, because biclustering cannot handle those. This left 6461 knees. The prepared dataset was then entered into the biclustering algorithm detailed below.

## Biclustering procedure and rationale

A natural way to identify candidate phenotypes of a disease from a dataset is to use clustering, an important type of unsupervised learning algorithm (i.e., there is no outcome variable), which assigns observations to groups according to a chosen measure of similarity. We sought to define knee phenotypes by clustering knees according to various measures in the OAI, such as demographics, clinical features, symptoms, and function. However, the OAI database contains many variables for each knee, many of which may be uninformative for that knee's phenotype. On one hand, such uninformative variables can hinder clustering algorithms from producing useful phenotypes, but on the other may provide crucial information for other phenotypes, and so cannot be completely discarded.

Biclustering, a procedure introduced to analyze gene expression data [22], can address this problem by clustering both observations and variables simultaneously. In contrast to standard clustering, which can be applied to either columns (i.e., clinical features) or rows (i.e., knees)

of a data matrix and produces unassociated subsets (the clusters), biclustering allows *simultaneous* clustering of both the rows and columns. It produces submatrices of the overall data matrix, which represent observations that are similar across a particular subset of the variables [25]. Since its introduction in 2000, there have been many modifications to biclustering for different applications [25, 26], but the original procedure [22] remains best suited to the current data, so this algorithm was applied without modification. In this paper, biclustering was used for phenotype identification, in the sense that knees of a given phenotype should be similar on clinical features associated with this particular phenotype, but allowed to be diverse on other variables. Thus, this two-dimensional algorithm will allow clustering of knees and clinical features simultaneously, capturing homogeneous subsets of knees with a coherent pattern across subsets of clinical features.

The quality of a bicluster may be measured by *mean square residue* (MSR) (16), which represents the average (squared) departure of the submatrix entries from their expected values for the bicluster. Cheng and Church's procedure [22] iteratively finds the largest biclusters (i.e., those with the most rows and columns) that achieve a MSR below a user-supplied threshold. The MSR threshold can vary from 0 to a maximum set by considering the entire dataset as a single large bicluster (in this case, 0.41). In the current analysis, an MSR threshold of 0.2 provided a balance between the size of the identified biclusters and strength of fit.

A high quality bicluster can also be viewed as a submatrix that is well-fit by two-way ANOVA [27], which can be represented by an $R^2$ value (i.e., the proportion of variance explained by membership in the bicluster) and is more readily interpreted than MSR.

## Statistical validation of biclusters

Clustering algorithms will readily identify candidate clusters in large datasets, but it is often unclear whether the identified clusters represent important underlying structure or are merely artifacts of natural sampling variation. Although several methods have been developed to address this statistical issue in biclustering [28–30], there are no methods that, to our knowledge, both align with our data setup and have procedures for determining statistical significance. Therefore, we additionally applied the SigClust procedure developed by Marron and colleagues for validating clusters [31]. SigClust tests whether clusters reflect a true separation in the data following a testing procedure. First, null Gaussian parameters are estimated using original OAI data to simulate *a proxy for unclustered data* from a single null Gaussian distribution. To test whether the data sample has stronger clusters than clusters from proxy data with a single cluster, the SigClust uses the *cluster index*, defined as the ratio of the within-cluster variation to the total variation. Tight, well-separated clusters have a low cluster index, while loose, overlapping clusters have a high cluster index. Since SigClust can only test two clusters at a time, we tested each pair of biclusters iteratively following the approach taken by Verhaak et al. [32]. SigClust produces a z-score quantifying the strength of evidence against the null hypothesis of a single Gaussian distribution (a proxy for unclustered data). Although z-scores can be converted to p-values (e.g., z-scores less than -2 correspond to one-sided p-values less than 0.023), we elected to present the z-scores to provide a better sense of the relative separation of various pairs of clusters in this exploratory analysis. In the following, a pair of clusters is considered "significantly different" when their SigClust z-score is less than -2.

## Preparation of the follow-up dataset

After identifying biclusters using only baseline data, we examined their relationships with outcome measures over 96 months of follow-up (Table 1). Kellgren Lawrence grade (KLG) progression and TKA receipt were based on available data. Calculation of quantitative joint space

**Table 1. Summary of outcomes over 96 months of follow-up in the OAI[*].**

| Outcome | Level | Description of each level |
|---|---|---|
| Incident or progressive radiographic OA (rOA) | 1 | Baseline KLG = 0–1, no follow up data (n = 0) |
| | 2 | Baseline KLG = 0–1, no incident rOA (n = 3127) |
| | 3 | Baseline KLG = 0–1, developed incident rOA (n = 557) |
| | 4 | Baseline KLG = 2+, no follow up data (n = 0) |
| | 5 | Baseline KLG = 2+, no progressive rOA (n = 2191) |
| | 6 | Baseline KLG = 2+, developed progressive rOA (n = 583) |
| | 7 | Baseline TKA (n = 0) |
| | 999 | No baseline and no followup KLG (n = 3) |
| Provision of total knee arthroplasty (TKA) | 0 | No TKA (n = 6060) |
| | 1 | Received TKA (n = 401) |
| Progressive structural damage, as quantitative joint space width [qJSW] loss[‡] | | % loss of qJSW[‡], in medial and lateral aspects |
| Progressive WOMAC[†] pain | 1 | Accelerated decrease in WOMAC pain from baseline |
| | 2 | No change in WOMAC pain over time |
| | 3 | Gradual worsening in WOMAC pain over time |
| | 4 | Increase then decrease in WOMAC pain |

[*] Available at: https://nda.nih.gov/oai/outcomes, *Outcome99*; KLG = Kellgren-Lawrence Grade;

[†] WOMAC: Western Ontario and McMaster Universities OA Index; Groups 1–4 defined by Group-based Trajectory modeling;

[‡] Loss of qJSW = difference in qJSW from baseline to current divided by baseline in mm, expressed as a percentage (continuous).

width (qJSW) loss and Western Ontario and McMaster Universities OA Index (WOMAC [33]) trajectories are described below.

**Quantitative joint space width loss.** QJSW is a continuous measure determined using an automated method as previously described [34]. We first fitted the linear regression models with qJSW at the medial (0.25 mm) and lateral (0.725 mm) locations as the dependent variables and time in months as an independent variable. Then, for each location, the fitted values from the regressions were used to define the progressive structural damage outcome as percentage decrease from baseline to 96 months (i.e., baseline minus current qJSW divided by baseline, to give percentage loss).

**WOMAC pain trajectories.** Group-based latent trajectory (GBLT) modeling was used to identify knees with different WOMAC pain trajectories, via PROC TRAJ (in SAS version 9.4), which fits a semiparametric mixture model to longitudinal data via maximum likelihood [35]. PROC TRAJ calculates the probability of each subject belonging to each latent trajectory group and assigns each subject to the group with the largest probability. We modeled the change from baseline (at 12, 24, 36, 48, 72, and 96 months) to reduce the effect of the variability in intercepts. The Bayesian Information Criterion (BIC) and conceptual clarity were used to determine the number of groups; shapes of the trajectories were determined based on the significance of polynomial terms. Due to attrition, especially after the 48th month, we included the dropout statement extension in the TRAJ procedure to account for missing data [36].

Table 2. Characteristics of included individuals from the OAI baseline cohort (n = 3330).

| Characteristics | Mean ± SD* or n (%) |
|---|---|
| Age, years (mean ± SD) | 61.4 ± 9.1 |
| BMI, kg/m² (mean ± SD) | 28.4 ± 4.7 |
| Women | 1910 (57.4%) |
| White | 2866 (86.1%) |
| Annual income > $50,000 | 2069 (64.0%) |
| Currently employed | 2073 (62.4%) |
| Incidence cohort | 2411 (72.4%) |
| Progression cohort | 895 (26.9%) |
| Healthy cohort | 24 (0.7%) |

*SD = standard deviation.

Joints with missing longitudinal data were included if they had the baseline measurement. The knees were classified according to a specific trajectory based on the maximum estimated probability of assignment. We considered a probability of 0.8 or higher as a good fit. Knees with estimated probability of 0.8 and higher were used to create bar plots showing the percentages of knees from a particular bicluster within the trajectory groups (the spaghetti plots of the raw trajectory data for the four groups are available in S1 Fig).

## Results

After exclusion of uninformative variables and features/knees with missing data, we included 86 features from 3330 people (6461 knees) for these analyses. Sample characteristics of participants who were included in the analysis are summarized in Table 2.

### Overall biclusters

We found 6 biclusters among the knees we analyzed (Table 3, Fig 1). These included biclusters with better (#1), similar (#2, 3, 6), and poorer (#4, 5) prognosis compared with the overall cohort (n = 6461) of knees. No knees were placed in more than one bicluster, and only 115 knees were not placed in a bicluster.

Bicluster #1 included 1425 knees from 960 individuals and 80 features, with an $R^2$ of 0.37 (Table 3). Knees in this bicluster had a lower frequency of TKR and progressive rOA compared with the full cohort and were less likely to develop rOA. These knees were less symptomatic,

Table 3. Summary of identified biclusters, number of knees and features included, and prognosis.

| Bicluster | n (knees) | N (people) | D (features) | $R^2$ | Prognosis* |
|---|---|---|---|---|---|
| 1 | 1425 | 960 | 80 | 0.37 | Better for bicluster |
| 2 | 2415 | 1649 | 69 | 0.30 | Similar |
| 3 | 1822 | 1294 | 63 | 0.31 | Similar |
| 4 | 188 | 147 | 70 | 0.64 | Poorer for bicluster |
| 5 | 238 | 191 | 64 | 0.53 | Poorer for bicluster |
| 6 | 258 | 194 | 63 | 0.31 | Similar |

*Summary prognosis for those knees in each bicluster compared to the full cohort of knees, based on outcomes shown in Table 1, including knees without radiographic OA (rOA), knees developing incident rOA (baseline KLG = 0–1 increasing to 2+), knees with prevalent rOA (baseline KLG = 2+ without progression), knees with progressive rOA (baseline KLG = 2+ with worsening KLG), and the proportion of knees undergoing TKR over 96 months.

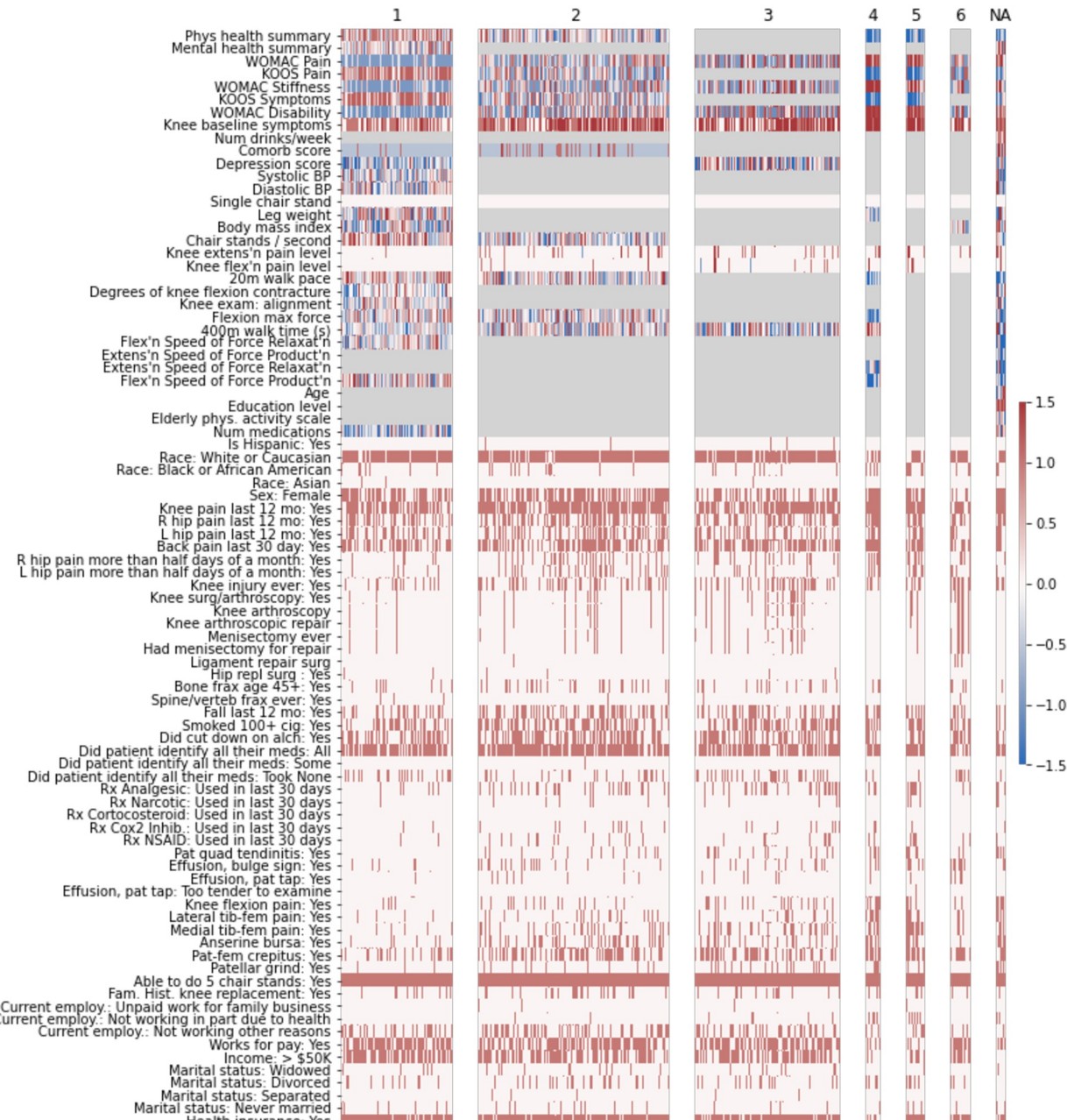

**Fig 1. Summary of 6 biclusters in overall data (all knees).** This heatmap shows all included variables on the y-axis, and knees on the x-axis. Biclusters are indicated across the top by number. The color scale indicates the direction of the variable on a standardized scale. Features/knees in gray are not included in the bicluster; na indicates that this group of knees were not included in any bicluster.

with higher KOOS pain and symptoms, and lower WOMAC scores (all scales). Knees in the bicluster were from individuals with better SF-12 physical summary scale scores and faster 20-meter walk speeds. Other features characterizing this bicluster included fewer depressive symptoms, fewer comorbidities, slightly higher income, and greater employment.

Biclusters #2, 3, and 6 had overall similar prognosis compared to the full cohort, but each included different knees and sets of features. The $R^2$ for these biclusters was around 0.3.

Bicluster #2 included 2415 knees (from 1649 individuals) and 69 features with minimal differences compared to the full cohort. Bicluster #3 included 1822 knees (from 1294 individuals) and 63 features. The knees in the bicluster, compared with the overall cohort, were from individuals with slightly more depressive symptoms and slightly slower 400-meter walk times. There was a higher frequency of prior injury in bicluster #2 compared with the full cohort, and knees were less often from women. Bicluster #6 included 258 knees from 194 people and 63 features. Included knees had lower WOMAC scores (all scales), higher KOOS pain scores, lower knee symptoms, and less patellar pain. The BMI of included participants was slightly lower than the full cohort. Similar to bicluster #3, there was a higher frequency of prior injury in the bicluster compared with the full cohort, and knees were less often from women.

Biclusters #4 and 5 were characterized by poorer outcomes. Bicluster #4 contained 188 knees (from 147 people) and 70 features and was characterized by greater frequency of both progressive rOA and TKA. Bicluster #4 also had the highest $R^2$ value of any of the 6 biclusters, at 0.64. Features that were higher among the knees in bicluster #4 versus the overall population of knees included knee symptoms, WOMAC scores, 400-meter walk time, knee tenderness on exam, patellar quadriceps tendinitis, inability to work for health reasons, presence of patellar grind, and knee flexion pain. In contrast, knees included in the bicluster had lower values for KOOS (to be expected as the coding is opposite that of WOMAC), SF-12 physical summary scale, 20-meter walk speed, and maximal isometric flexion strength (Fig 2). These knees were less likely to be from Hispanic or Asian participants, individuals who were separated (versus any other marital status), and less frequently reported intra-articular corticosteroids in the last 30 days.

Similarly, bicluster #5 included 238 knees from 191 people and 64 features, with an $R^2$ of 0.53. Knees included in this bicluster were more likely to have knee rOA at baseline, and more likely to experience radiographic progression, as well as to undergo TKA over the course of the study. Knees included in the bicluster were characterized by higher WOMAC scores (all scales), lower KOOS pain and symptoms, more knee symptoms, lower SF-12 physical summary scale scores, and greater pain with knee flexion and extension. Included knees were from individuals who were more likely not working due to health issues, more likely using narcotics (and medications in general), more frequently Black, and less likely to have income over $50,000 per year.

## Bicluster comparison by baseline measures

To compare biclusters on key variables, we generated a boxplot for continuous variables (Fig 3) and a bar plot for categorical variables (Fig 4). As an example, the leftmost set of boxplots describes the distribution of Physical Health Summary within each bicluster. The variables have been standardized by the procedure described above (page 5), such that 0 denotes the sample mean and the box in each boxplot marks the middle 50% of the values. The interquartile range (IQR) for bicluster #1 (red) stretches from about 0 to 1, indicating that the middle 50% of knees in bicluster #1 have Physical Health Summary values between 0 and 1 standard deviations above the sample mean. Biclusters #4 (cyan) and #5 (blue) have their middle 50% of knees between 0.5 and 1.5 standard deviations *below* the mean. Biclusters 3 and 6 do not incorporate the Physical Health Summary variable, so there is no boxplot for those biclusters.

The U-shaped pattern (over biclusters) observed for Physical Health Summary is also noted in the KOOS and WOMAC variables: bicluster #1 tends to be the least symptomatic/impaired, while biclusters #4 and 5 are the most symptomatic or impaired in these variables. Unlike the physical health summary and KOOS variables, the WOMAC variables use high numbers to indicate worse health, so we plotted the negative standardized values of the WOMAC variables

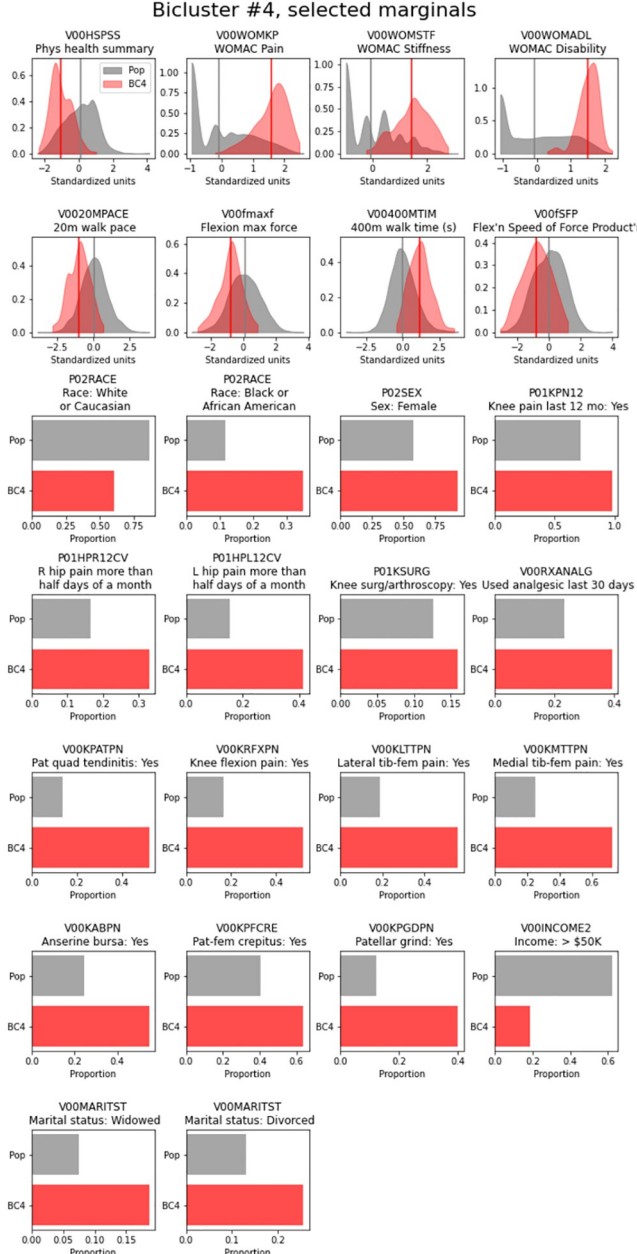

**Fig 2. Distribution of top features for bicluster 4 (all knees).** Marginal plots of key features for bicluster 4 are shown, with the distribution for the overall cohort of knees in gray and the distribution of each feature in bicluster #4 shown in red; vertical lines indicate the mean value for each group.

to visualize this pattern. This pattern is also partially visible in the variables on the right side of the visualization: bicluster #1 shows faster, stronger knees in the 400m walk time, 20m walk pace, and force variables, bicluster #2 reflects the cohort, and bicluster #4 has slower, weaker knees. Biclusters #5 and 6 did not contain these variables.

Among binary variables (Fig 4), similar patterns are seen, where bicluster #1 is characterized by less pain and higher socioeconomic status (SES), bicluster #2 reflects the cohort, and biclusters #4 and 5 demonstrate more pain and lower SES. The binary variables are not

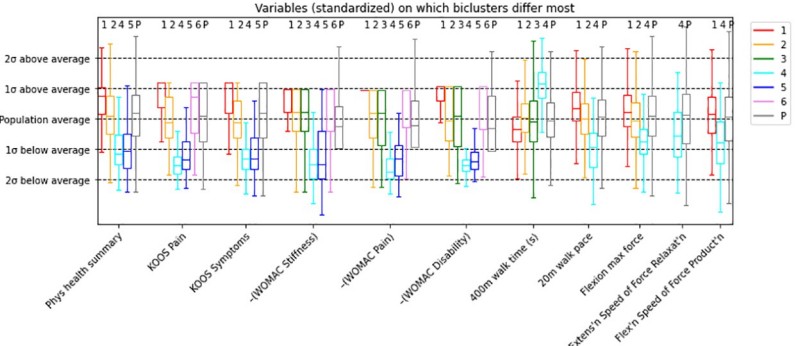

**Fig 3. Boxplot comparing key continuous variables across the 6 biclusters and the overall population of knees (P).** The y-axis represents standardized values and standard deviations, and the x-axis indicates the features. Boxplots are colored by bicluster numbers, with P = population in gray indicating the overall cohort mean and distribution.

standardized (e.g., the value of the red bar in the *Works for pay* variable is about 0.7 and indicates that about 70% of observations in bicluster #1 work for pay). The grouped bars in this plot form three basic shapes: a U shape in the first three variables, a mound shape in variables from *Bone frax age 45+* (i.e., bone fracture over age 45 years) to *Back pain last 30 days*, and a much sharper mound shape for the variables from *Race: Black or African American* to *Pat-fem crepitus* (i.e., patellofemoral crepitus). This last set of variables indicates relatively large differences between bicluster #1 and biclusters #4 and 5, while the remaining biclusters reflect the overall cohort.

## Bicluster validation using SigClust

Pairwise SigClust using all features demonstrated a significant difference between biclusters #1 and 4 with a z-score of -10.2, between biclusters #1 and 5 with a z-score of -7.4, and between

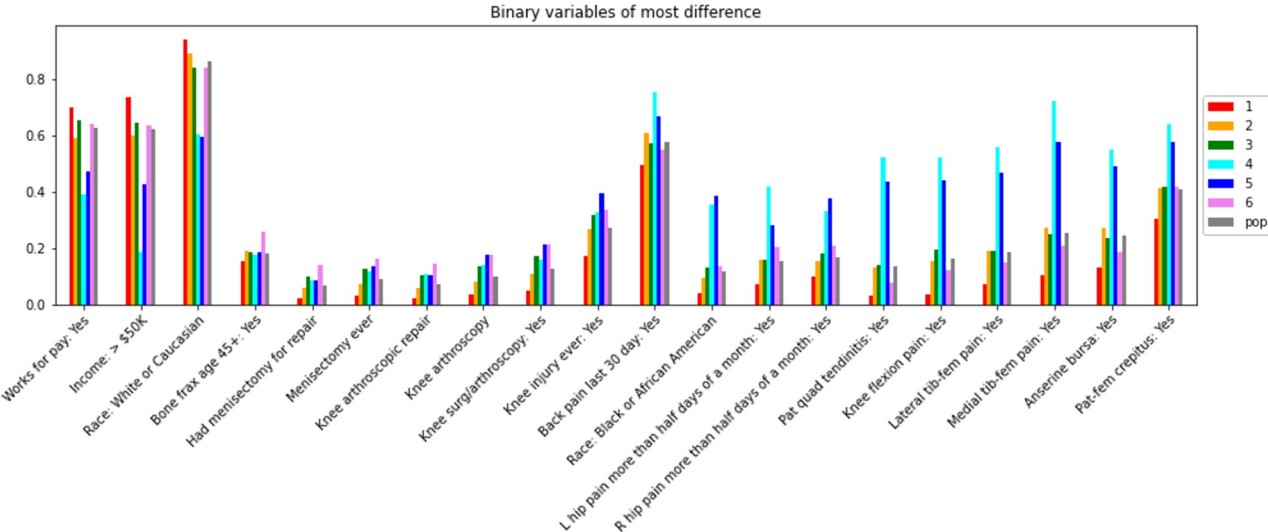

**Fig 4. Bar plot comparing key binary variables across the 6 biclusters and the overall population of knees (pop).** The y-axis indicates the proportion in each category of the variables (which are shown on the x-axis). Bars are colored by bicluster number, with the overall population of knees (pop) in gray.

**Fig 5. Pairwise SigClust among each pair of biclusters.** Each pair of biclusters is plotted in a two-dimensional scatterplot: the horizontal axis of each scatterplot is the mean-difference direction (the vector between the centers of the biclusters) and the vertical axis is the orthogonal principal component to the mean-difference direction. The SigClust z-score for the pair of biclusters is printed on the plot in black, and in red if it is below -2 (which would demonstrate statistical significance in the absence of multiple comparisons).

biclusters #4 and 6 with a z-score of -3.6. Z-scores between other pairs of biclusters did not demonstrate statistical significance (Fig 5).

## Comparison of biclusters by outcomes

As noted in Table 3, the outcome measures vary substantially for the individual biclusters. This is demonstrated for radiographic progression (i.e., development or worsening of radiographic OA by Kellgren-Lawrence Grade [KLG]) in Fig 6A and for receipt of TKA by 96 months in Fig 6B.

(A)

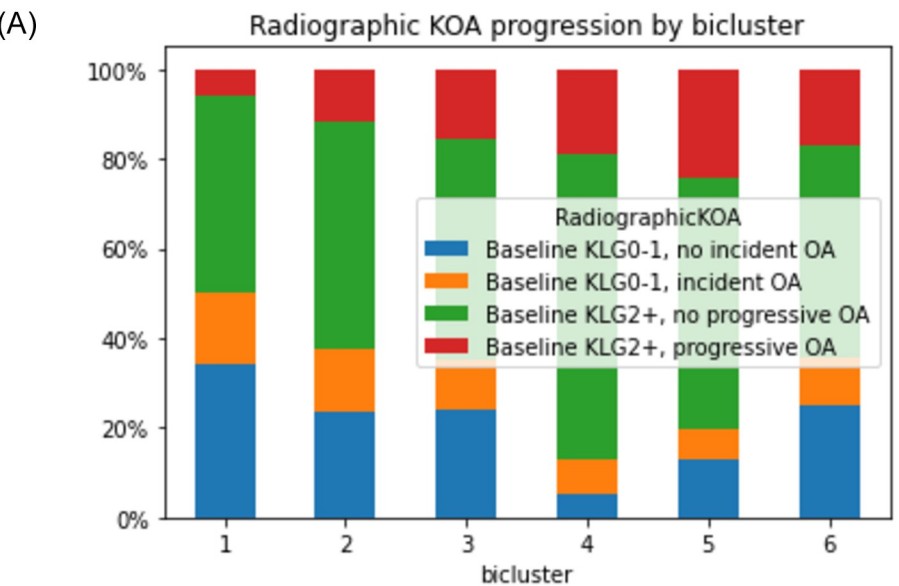

(B)

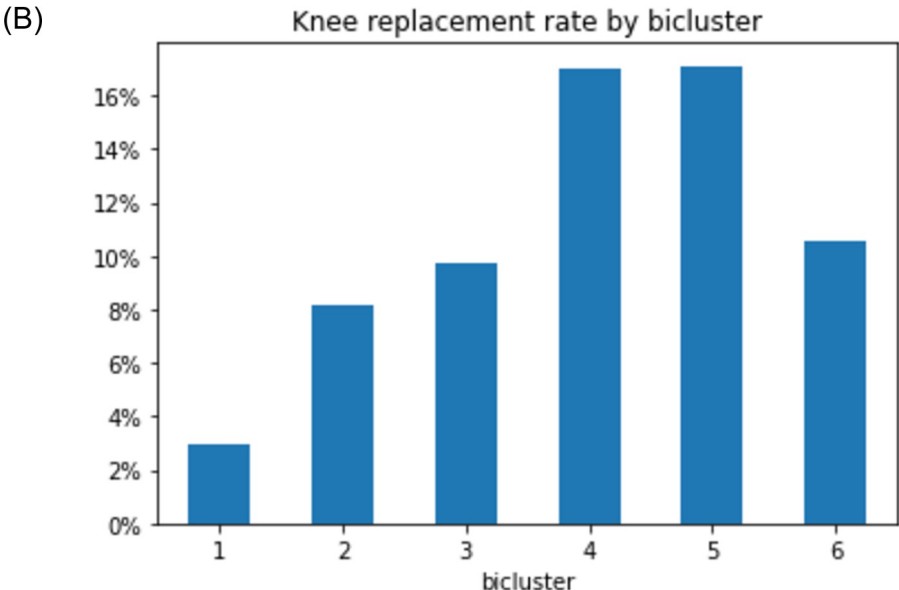

**Fig 6. Structural progression by bicluster over 96 months. A.** Progression based on Kellgren-Lawrence Grade (KLG) from baseline to 96 months; **B.** Progression based on receipt of a total knee arthroplasty (TKA) by 96 months. The x-axis indicates each of the six biclusters; the y-axis is the proportion of knees in each category of the legend.

Incident and worsening rOA are summarized together in Fig 6A, where knees in blue do not have rOA at baseline and do not develop it over the study period; green indicates rOA at baseline without progression over the study period. Orange indicates development of new knee rOA, while red reflects progression of rOA over 96 months. The x-axis shows the 6 biclusters; knees in bicluster #4 and 5 have more prevalent and progressive rOA compared to the other biclusters. Knees without baseline or new rOA (blue) are more frequent in biclusters #1–3 and 6. Similarly, biclusters #4 and 5 had the most eventual TKA over the 96 months at around 16% (Fig 6B), while bicluster 1 had the fewest at only about 3%. Biclusters #2, 3, and 6 had rates between 8–10%.

(A)

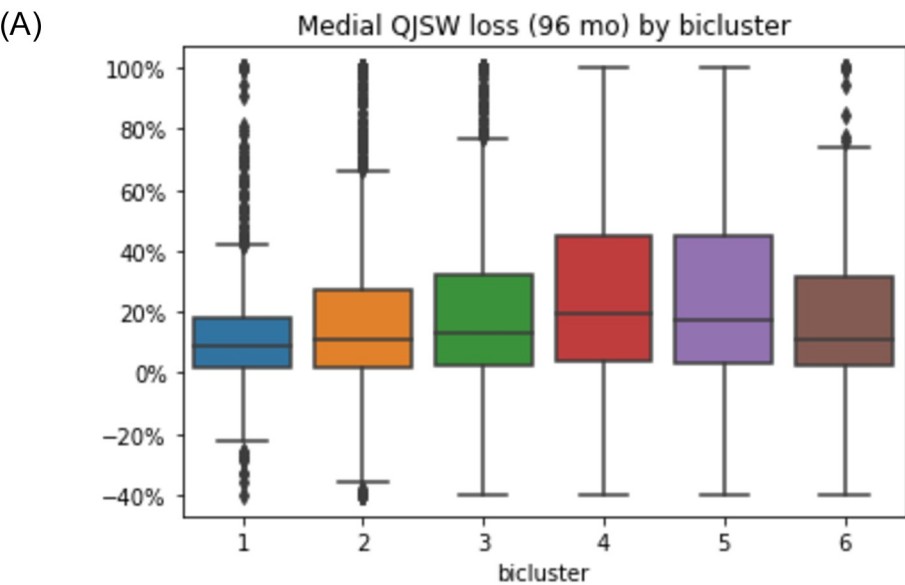

(B)

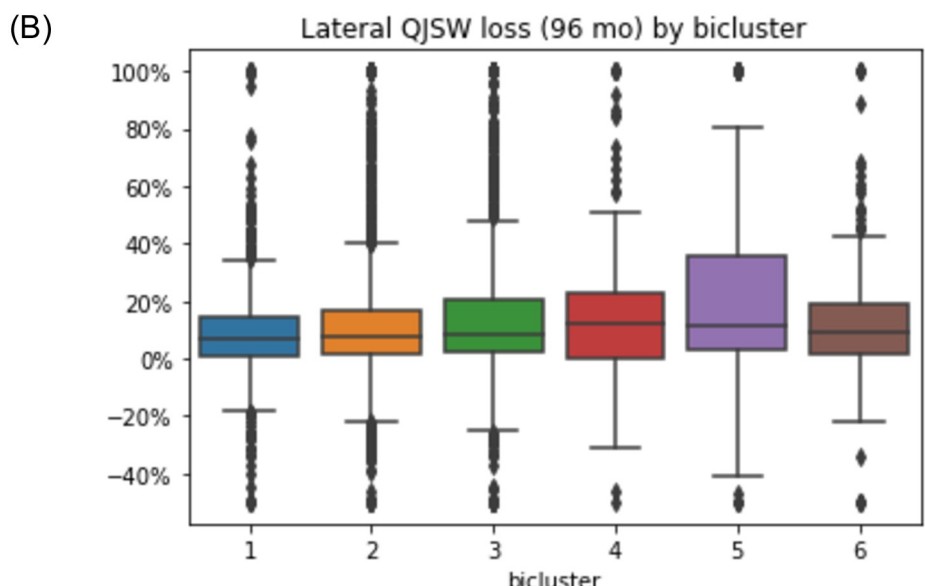

**Fig 7. Percentage loss in quantitative joint space width (qJSW) in the medial (A) and lateral (B) compartments for each of the six biclusters on the x-axis.**

When progression was assessed using medial compartment qJSW (Fig 7A), a similar pattern was identified, with biclusters #4 and 5 demonstrating more medial joint space loss, around 20%, while bicluster #1 had the least loss, with a median loss of less than 10% of the baseline width. In contrast, less difference in the median % loss was noted in the lateral compartment among the biclusters (Fig 7B), although bicluster #5 was noted to have a larger number of knees losing a higher percentage of lateral qJSW. Estimated medial qJSW loss at 96 months was truncated at 100% when the linear regression resulted in loss >100% (3% of knees), or if gain was >40% (1% of knees); for lateral qJSW loss, the thresholds were >100% and gain of >50% (1.6% and 1.2% of knees, respectively).

**Comparison of biclusters based on pain outcome.** Using GBLT modeling, we identified four WOMAC pain trajectories over 96 months (Fig 8A). Most knees were on the "stable" trajectory (Group 2), characterized by no change in WOMAC pain over time. Probability of membership in this group was estimated at 76.4% (95% CI [75.3, 77.5]). Fewer knees (11.2%, 95% CI: [10.5, 11.9]) were on the "improvement" trajectory (Group 1), initially experiencing more pain, with accelerated decrease over time. Similarly, 9% (95% CI: [8.0, 10.1]) of knees were on a "worsening" trajectory (Group 3) with an overall steady increase in pain. The smallest group (3.3%, 95%CI: [2.7, 4.0]) of knees were on an irregular trajectory (Group 4), experiencing an increase in pain from baseline followed by a decrease to roughly baseline level.

After defining the trajectory groups, we examined how they were represented within each bicluster (Fig 8B). We found that knees in bicluster #4 and 5 contained relatively more knees of Group 1 (improvement) than other biclusters. In contrast, knees in bicluster #1 were more likely to demonstrate stable pain over time (95% of knees in bicluster #1 were assigned to the stable trajectory, Group 2). Only 4 to 7% of the knees in each bicluster demonstrated a worsening pain trajectory over time (Group 3), and even fewer (1–4% in each bicluster) were in Group 4, with initial worsening and later improvement. Although infrequent, these latter two patterns may be of the greatest interest regarding a future clinical trial targeting these knees as having the greatest potential for symptomatic improvement.

## Discussion

We identified six distinct, non-overlapping biclusters among more than 6000 knees in more than 3000 OAI participants when examining 86 different features (Table 4). Of these, bicluster #1 represented a group of generally healthier knees with a lower frequency of OA development and progression; biclusters #4 and 5 were the least healthy, with more frequent OA and greater progression, including a greater frequency of TKA. Biclusters #4 and 5 were statistically distinct from bicluster #1 according to SigClust. The patterns of most features were similar for biclusters #4 and 5, reflective of greater symptoms and tenderness, poorer physical health, and slower walking speeds. Contrasting features included reduced isometric flexion strength and less intra-articular corticosteroid use in bicluster #4, while knees included in bicluster #5 were more often from Black individuals, those taking narcotics, and with lower incomes. Interestingly, these groups also differed somewhat based on qJSW loss, with more medial qJSW loss for bicluster #4 and more lateral qJSW loss in bicluster #5.

Our work supports previous research identifying a variety of features associated with KOA progression, such as more symptoms/tenderness in the knee and other joints, female sex, non-white race, low income, lack of employment, depression/comorbid conditions, obesity, and knee injury [37–39]. However, the present work has the added value of finding and quantifying the associations of all these features, in subsets of knees, in a single comprehensive analysis and choosing the key differentiating features in a data-driven manner.

Some recently published studies highlight the breadth of approaches possible with advanced methodologies and the rich OAI database, mostly focused on prediction models. For example, Guan, et al. [40] utilized deep learning (DL) to predict progression of pain (yes/no based on 9-point increase in WOMAC pain) from a baseline knee radiograph, finding that the DL model alone had a predictive area under the curve (AUC) of 0.77 which increased to 0.81 after addition of age, sex, race, BMI, WOMAC pain, and KLG to the model; prediction was better when restricted to knees with baseline rOA. Other groups have sought to predict structural progression with machine learning models. In one of several works in this area, Jamshidi et al. [15] used baseline OAI features, including quantitative MRI, to predict cartilage volume loss and radiographic medial joint space narrowing, finding AUCs of 0.7 to 0.9 for these

(A)

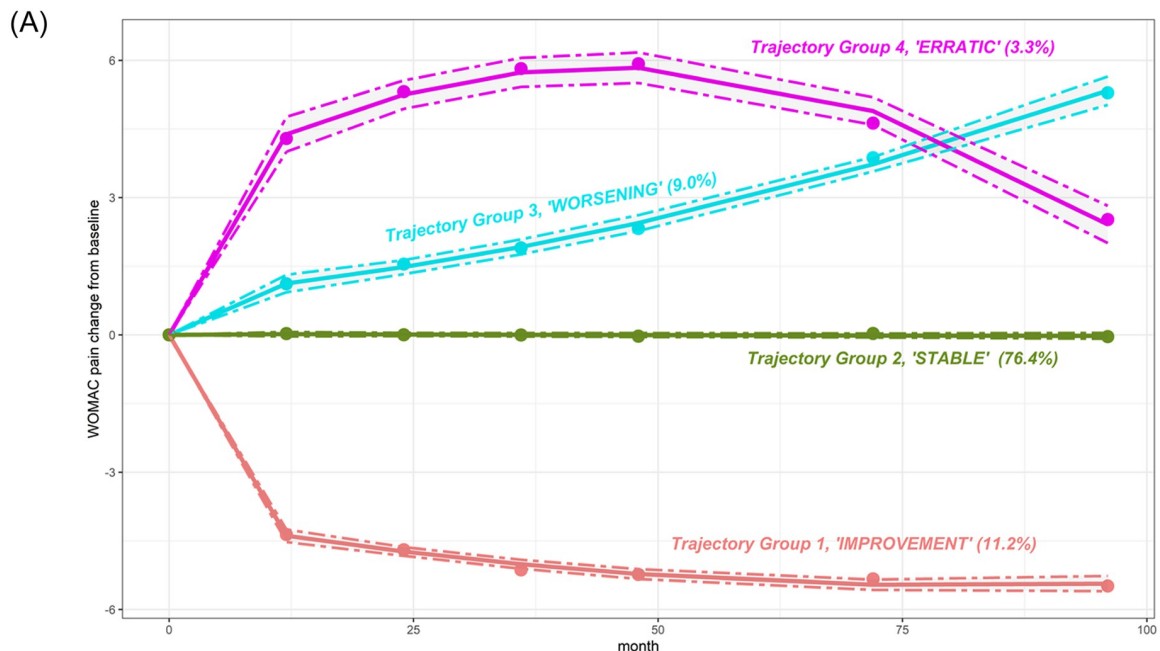

(B)

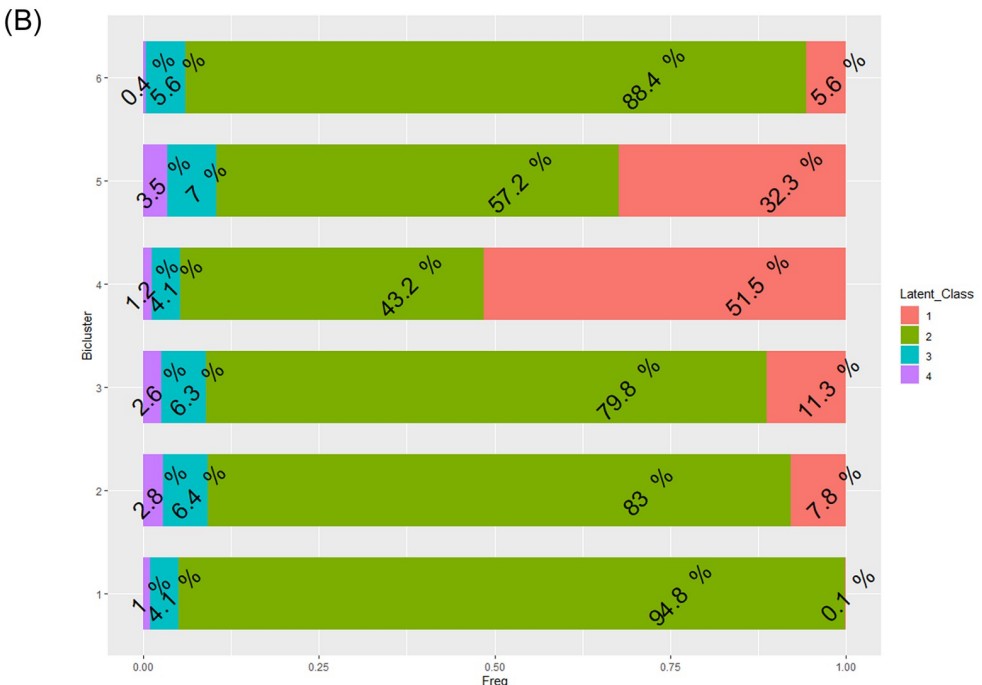

**Fig 8.** **A.** Estimated trajectories of WOMAC pain change with 95% confidence intervals (dotted lines) defined by GBTM. Group 1 (red): 11.2% of knees with a mean probability of assignment of 0.94±0.12. Group 2 (green): 76.4% of knees with a mean probability of assignment of 0.95±0.10. Group 3 (cyan): 9.0% of knees with a mean probability of assignment of 0.83±0.16. Group 4 (pink): 3.3% of knees with a mean probability of assignment of 0.88±0.15. See also S1 Fig. **B.** Assignment to the trajectory by bicluster. Each horizontal bar represents a bicluster. The numbers within the segments of the bar are proportions of knees assigned to a specific trajectory.

Table 4. Summary of outcomes by bicluster.

| Bicluster # | Radiographic OA | Arthroplasty | Medial narrowing* | Lateral narrowing* | Symptoms |
|---|---|---|---|---|---|
| | *KLG* | *TKR* | *medial qJSW* | *lateral qJSW* | *WOMAC* |
| 1 | **Less rOA at baseline and less progression** | **3% TKR** | *2–18% loss* | *1–15% loss* | **95% "stable"** |
| | | | | | **0.1% "improver"** |
| 2 | *Like population* | *8% TKR* | *2–28% loss* | *1–17% loss* | *83% "stable"* |
| 3 | *Like population* | *10% TKR* | *2–32% loss* | *2–21% loss* | *80% "stable"* |
| 4 | **Most rOA at baseline, more progression** | **17% TKR** | **4–45% loss** | *0–23% loss* | **52% "improvers"** |
| | | | | | **4% "worsening"** |
| 5 | **More rOA at baseline, most progression** | **17% TKR** | **3–44% loss** | **3–36% loss** | **32% "improvers"** |
| | | | | | **7% "worsening"** |
| 6 | *Like population* | *11% TKR* | *2–31% loss* | *1–17% loss* | *88% "stable"* |

*25th-75th percentile (IQR); KLG = Kellgren Lawrence Grade; TKR = total knee replacement; qJSW = quantitative joint space width; WOMAC = Western Ontario and McMaster University Osteoarthritis Index.

relationships. These investigators also explored baseline biomarkers as predictors of KOA progressors [41].

Our work contrasts with prior studies as we used an unsupervised approach, rather than a supervised one. Therefore, we did not attempt to predict progression, but rather to identify subgroups within the baseline data and subsequently compare these groups on key outcomes (including progression). A key strength of the biclustering approach is that it allows simultaneous clustering of both observations (e.g., knees) and features, as some features are only informative for some phenotypes. The presence of variables in the dataset, that are informative for some phenotypes but not others, is not unique to the OAI and is becoming more common as new technologies have enabled storing, organizing, and computing with large amounts of data. Biclustering accounts for this issue and may provide even more value when we integrate additional data types, such as MRI scores or cartilage maps, biomarkers, etc., that may specifically relate to only some phenotypes and uninformative for others.

While each knee was only included in one bicluster, and only a few knees were not included in a bicluster, there were similarities among individuals and among features. This highlights the significant overlap among phenotypes in knee OA [11], and the lack of clearly separated clusters as have been seen in cancer [42, 43] and other fields [44]. Of particular interest is a recent work using RNAseq and multi-omics factor analysis on data from cartilage and synovial samples of patients undergoing joint replacement [45]. This study identified distinct and independent subgroups among tissue from low-grade cartilage lesions and from synovial tissue, all of which were characterized by variations in inflammation, extracellular matrix, and cell adhesion. No subgroups were seen in tissue from the more severely damaged high grade cartilage lesions. Their analysis suggested that the variation among groups was along a continuum, the "inflammatory endotype axis of variation," rather than reflecting discrete clusters. This conclusion is in line with our work to date, suggesting that subgroups of OA may be more subtle, and less discrete, than have been found in other disease processes. The identified biclusters provide a starting point for further work to explore the underpinnings of each, and further explore potential key features to target for interventions (i.e., in biclusters #4 and 5), to exclude in clinical trials of progression or to focus on for prevention trials (i.e., features of bicluster #1).

### Strengths and limitations

We have demonstrated the ability to identify phenotypes in the baseline visit of the OAI that reflect different prognosis over longitudinal follow up. The main strength of this work is in demonstrating that the central ideas of biclustering are useful for OA subtyping. Our work is the first application of biclustering in OA, and the first usage of SigClust for testing the significance of biclustering results in any field.

Confirmatory analysis applying these biclusters to an independent dataset in a supervised approach could support the importance of these exploratory findings. A re-design of the biclustering algorithm could better reflect the structure of this and other clinical datasets in the future. Additionally, to avoid the extra step of determining significance, if the statistical significance step could be directly incorporated into, and optimized for, the biclustering process, it may result in higher quality biclusters and a streamlined process.

### Conclusions

We identified six biclusters (groups of features and knees) within the baseline OAI data with varying prognoses. Such biclusters may represent potential KOA phenotypes (e.g., progressor phenotype(s)) within the larger cohort. Novel application of existing methodologies can provide insights into OA phenotypes and development or progression of disease. Additionally, the identification of phenotypes with differing prognostic associations may identify groups that are most likely to respond to specific interventions.

### Supporting information

**S1 Table. OAI variable descriptions and key to NDA dataset.**
(PDF)

**S1 Fig. Spaghetti plots.**
(PDF)

### Acknowledgments

We would like to thank the Thurston Arthritis Research Center and the Core Center for Clinical Research for valuable feedback regarding this work.

### Author Contributions

**Conceptualization:** Amanda E. Nelson, Liubov Arbeeva, J. S. Marron.

**Data curation:** Liubov Arbeeva.

**Formal analysis:** Thomas H. Keefe, Liubov Arbeeva, J. S. Marron.

**Funding acquisition:** Amanda E. Nelson, Leigh F. Callahan, Richard F. Loeser.

**Investigation:** Amanda E. Nelson, Thomas H. Keefe, Todd A. Schwartz, Yvonne M. Golightly, Liubov Arbeeva, J. S. Marron.

**Methodology:** Thomas H. Keefe, Todd A. Schwartz, Liubov Arbeeva, J. S. Marron.

**Project administration:** Amanda E. Nelson, Yvonne M. Golightly.

**Resources:** Amanda E. Nelson, Liubov Arbeeva.

**Software:** Thomas H. Keefe, Liubov Arbeeva, J. S. Marron.

**Supervision:** Amanda E. Nelson, Todd A. Schwartz, Leigh F. Callahan, Richard F. Loeser, Yvonne M. Golightly, J. S. Marron.

**Validation:** Thomas H. Keefe, Liubov Arbeeva, J. S. Marron.

**Visualization:** Amanda E. Nelson, Thomas H. Keefe, Liubov Arbeeva, J. S. Marron.

**Writing – original draft:** Amanda E. Nelson, Thomas H. Keefe, Liubov Arbeeva.

**Writing – review & editing:** Amanda E. Nelson, Thomas H. Keefe, Todd A. Schwartz, Leigh F. Callahan, Richard F. Loeser, Yvonne M. Golightly, Liubov Arbeeva, J. S. Marron.

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
