## [Decision Letter · Decision Letter 0]

17 Feb 2022

PONE-D-21-36664Biclustering reveals potential knee OA phenotypes in exploratory analyses: Data from the Osteoarthritis InitiativePLOS ONE

Dear Dr. Nelson,

Thank you for submitting your manuscript to PLOS ONE. After careful consideration, we feel that it has merit but does not fully meet PLOS ONE’s publication criteria as it currently stands. Therefore, we invite you to submit a revised version of the manuscript that addresses the points raised during the review process.

We look forward to receiving your revised manuscript.

Kind regards,

Shazlin Shaharudin

Academic Editor

PLOS ONE

Journal Requirements:

Reviewers' comments:

Reviewer's Responses to Questions

**Comments to the Author**

1. Is the manuscript technically sound, and do the data support the conclusions?

Reviewer #1: Yes

Reviewer #2: Yes

Reviewer #3: Partly

2. Has the statistical analysis been performed appropriately and rigorously? 

Reviewer #1: Yes

Reviewer #2: Yes

Reviewer #3: Yes

3. Have the authors made all data underlying the findings in their manuscript fully available?

Reviewer #1: Yes

Reviewer #2: Yes

Reviewer #3: Yes

4. Is the manuscript presented in an intelligible fashion and written in standard English?

Reviewer #1: Yes

Reviewer #2: Yes

Reviewer #3: Yes

5. Review Comments to the Author

Reviewer #1: The paper provides very good data for study related to OA. I do not think that there are concerns about dual publication, research ethics, or publication ethics. I am positive that this study will provide good data support for future research work.

Reviewer #2: This paper proposed biclustering, which is simultaneously performed on both cluster observations and clinical features, to explore candidate phenotypes of knee osteoarthritis (KOA) within the Osteoarthritis Initiative (OAI) dataset. The topic is interesting and valuable for the research community, and the manuscript can be improved by providing more details in the “Methodology” section.

In “Data preparation”, marginal distribution plots are not provided in this paper. Are these plots strongly violate the approximate normality assumption? I think this may be the motivation for using the data transformation method in [23]. More importantly, I suggest the authors justify why the knee-based approach is better than the person-based approach. In addition, what are the criteria of the data cleaning process? e.g., how to select 86 features among 116 clinical and demographic features? What kind of feature selection techniques is used?

The two essential parts of biclustering are the good measure and the suitable heuristics search method [29]. The details of the algorithm can make the manuscript more convince.

In “Strengths and Limitations”, the authors claimed that the method proposed in this paper is a novel two-step process. However, validation is a common step in any machine learning algorithm. Therefore, the authors need to update the contribution instead of overclaiming. The same problem happened in “Conclusion”, the method proposed in this paper is not new.

Minor comments:

• The “Definitions and levels” are not clear in Table 1. Are the second column definitions and third column levels? Moreover, what does “% loss of qJSW” mean?

• In S1 Figures, what is the unit of the time t (month?). In addition, these plots missed legends for different trajectories.

• Reference [16] has the format problem in page 7, section “Biclustering procedure and rationale”.

Reviewer #3: Abstract

-what has been done in the past similar studies?

-to mention the problem statement/research gap

Introduction

-mention the limitation of existing works

-is biclustering a novel approach for KOA area?

Methodology

-please detail the variables considered in the study. Those variables suddenly appear in Results sections & Figures.

e.g. 400m walk time, 20m walk pace, and force variables?

-suggest to change subheading “Outcomes” to “Analysis” instead to avoid confusion with the Results (Outcomes). Also the same word in Table 1 caption.

-subtitle “JSW loss” same as “qJSW loss”. Confusing. Please clarify.

-to detail how statistical significance is determined.

Biclustering procedure and rationale section:

-“However, the OAI database contains many variables for each knee, many of which may be uninformative for that knee’s phenotype. On one hand, such uninformative variables can hinder clustering algorithms from producing useful phenotypes, but on the other may provide crucial information for other phenotypes, and so cannot be completely discarded.”

This should be under Data Preparation/Data Cleaning stage instead.

-The writings is more towards what is biclustering, the background of biclustering and literature works instead of the authors’ actual research methodology. Please rewrite to reflect how it is applied in the case study.

Results

-please explain why only women included in the case study.

-Figures 1, 2, 5 are blur

Bicluster comparison by baseline

-Please label or indicate where is the baseline in the boxplot

e.g. “As an example, the leftmost set of boxplots shows that bicluster #1 (red) has above average values for Physical Health Summary, while 4 (cyan) and 5 (blue) have below average values.”

Where are the average values? How much?

-entire manuscript is not well written specifically the methodology & results section

-hard to relate the process mentioned in methodology to the results.

Discussion, Strength & Limitations

-These sections are well written.

Format

-mistake in citation style. Please check all.

e.g. “For example, Guan, et al utilized deep learning (DL)..

6. PLOS authors have the option to publish the peer review history of their article (what does this mean?). If published, this will include your full peer review and any attached files.

Reviewer #1: No

Reviewer #2: **Yes: **Huanghe Zhang

Reviewer #3: No

---

## [Author Response · Author response to Decision Letter 0]

28 Mar 2022

Please see attachment for formatted version of response.

RESPONSE TO REVIEWERS: PONE-D-21-36664

Reviewer #1: The paper provides very good data for study related to OA. I do not think that there are concerns about dual publication, research ethics, or publication ethics. I am positive that this study will provide good data support for future research work.

Author response: Thank you very much for your time and positive comments.

Reviewer #2: This paper proposed biclustering, which is simultaneously performed on both cluster observations and clinical features, to explore candidate phenotypes of knee osteoarthritis (KOA) within the Osteoarthritis Initiative (OAI) dataset. The topic is interesting and valuable for the research community, and the manuscript can be improved by providing more details in the “Methodology” section.

Author response: Thank you for your time and constructive comments, addressed below.

R2Q1: In “Data preparation”, marginal distribution plots are not provided in this paper. Are these plots strongly violate the approximate normality assumption? I think this may be the motivation for using the data transformation method in [23]. More importantly, I suggest the authors justify why the knee-based approach is better than the person-based approach. In addition, what are the criteria of the data cleaning process? e.g., how to select 86 features among 116 clinical and demographic features? What kind of feature selection techniques is used?

Author response/changes made: Thank you for this observation. We have added further detail regarding the approach, specifically the transformation process and variable selection criteria, under a new subheading “Preparation of the baseline dataset” on p4-5. We did not include the 172 marginal distribution plots (86 features before and after transformation) due to space and lack of additional relevant information for this analysis. See also R3Q3-4.

R2Q2: The two essential parts of biclustering are the good measure and the suitable heuristics search method [29]. The details of the algorithm can make the manuscript more convince.

Author response/changes made: In “Biclustering Procedure and Rationale”, we clarified that we used the Cheng & Church algorithm without modification, available in reference #22 (p6).

R2Q3: In “Strengths and Limitations”, the authors claimed that the method proposed in this paper is a novel two-step process. However, validation is a common step in any machine learning algorithm. Therefore, the authors need to update the contribution instead of overclaiming. The same problem happened in “Conclusion”, the method proposed in this paper is not new.

Author response/changes made: Thank you for pointing this out and allowing us to clarify one of the key strengths of this work. We have edited this sentence (p20) to read “Our work is the first application of biclustering in OA, and the first usage of SigClust for testing the significance of biclustering results in any field.” In conclusion, we changed the sentence as follows: “Novel application of existing methodologies can provide insights into OA phenotypes and development or progression of disease.”

R2Q4: The “Definitions and levels” are not clear in Table 1. Are the second column definitions and third column levels? Moreover, what does “% loss of qJSW” mean? In S1 Figures, what is the unit of the time t (month?). In addition, these plots missed legends for different trajectories.

Reference [16] has the format problem in page 7, section “Biclustering procedure and rationale”.

Author response/changes made: We have reformatted Table 1 to clarify the columns and moved the description of qJSW loss to a footnote in this table. The S1 Figures were replotted with the additional information. I am not sure what formatting issue the reviewer is referring to, but we will be happy to address anything as needed per the publisher’s requirements.

Reviewer #3: 

R3Q1: Abstract, what has been done in the past similar studies? to mention the problem statement/research gap

Author response: No similar work has been done in OA. There are studies of machine learning as noted in the introduction, but there is not enough room in the abstract to expand upon this. 

Changes made: Added “for the first time” to the objective statement in the abstract.

R3Q2: Introduction, mention the limitation of existing works, is biclustering a novel approach for KOA area?

Author response: See also R2Q3 and R3Q1. We added to the discussion a sentence as follows (p20) “Our work is the first application of biclustering in OA, and the first usage of SigClust for testing the significance of biclustering results in any field.”

R3Q3: Methodology, please detail the variables considered in the study. Those variables suddenly appear in Results sections & Figures (e.g., 400m walk time, 20m walk pace, and force variables). Suggest to change subheading “Outcomes” to “Analysis” instead to avoid confusion with the Results (Outcomes). Also the same word in Table 1 caption. Subtitle “JSW loss” same as “qJSW loss”. Confusing. Please clarify. To detail how statistical significance is determined.

Author response: As we note in the Methodology/Data Source and Data reporting section, All OAI data used in this paper are publicly available. Data used for these analyses are available at https://nda.nih.gov/study.html?id=911; DOI: 10.15154/1519056. Additionally, all variables are detailed in the Supplemental Table. Extensive detail about the variables is widely available in other OAI publications and on the website referenced in the paper, and as this is not the point of the present work, we did not repeat this text here. 

Changes made: We changed and re-ordered the subheadings under Methodology to: Data source; Preparation of the baseline dataset; Biclustering procedure and rationale; Statistical validation of biclusters; Preparation of the follow-up dataset, and subheadings of Quantitative joint space width loss and WOMAC pain trajectories. The section on statistical validation of biclusters was moved under the section on Biclustering procedure in the methods.

R3Q4: Biclustering procedure and rationale section: “However, the OAI database contains many variables for each knee, many of which may be uninformative for that knee’s phenotype. On one hand, such uninformative variables can hinder clustering algorithms from producing useful phenotypes, but on the other may provide crucial information for other phenotypes, and so cannot be completely discarded.” This should be under Data Preparation/Data Cleaning stage instead.

Author response/changes made: We have added additional detail as recommended under the new “Preparation of the baseline dataset” section starting on page 4. 

R3Q5: The writings is more towards what is biclustering, the background of biclustering and literature works instead of the authors’ actual research methodology. Please rewrite to reflect how it is applied in the case study.

Author response/changes made: Please see also R2Q2 regarding the algorithm. In the 2nd paragraph under “Biclustering procedure and rationale,” on page 6, we added the following: “In this paper, biclustering was used for phenotype identification, in the sense that knees of a given phenotype should be similar on clinical features associated with this particular phenotype, but allowed to be diverse on other variables. Thus, this two-dimensional algorithm will allow clustering of knees and clinical features simultaneously, capturing homogeneous subsets of knees with a coherent pattern across subsets of clinical features.”

R3Q6: Results: Please explain why only women included in the case study.

Author response: The OAI, and the present analysis, include men and women. There were 57% women as noted in the abstract, leaving 43% men of the 3300 individuals included in these analyses. These numbers are also provided in Table 2. We only listed the % women in the table since readers would understand the remainder are men.

R3Q7: Figures 1, 2, 5 are blur. Please label or indicate where is the baseline in the boxplot

e.g. “As an example, the leftmost set of boxplots shows that bicluster #1 (red) has above average values for Physical Health Summary, while 4 (cyan) and 5 (blue) have below average values.” Where are the average values? How much?

Author response: Thank you for calling our attention to this. We used the image preparation tool recommended by the journal and will work with them to optimize the figures before publication. Regarding the boxplot (Figure 3), we have clarified in the title that these are standardized variables and added an indication on the y-axis regarding the number of standard deviations above or below the sample average the bicluster values were. We updated the description of Figure 3 to match (page 14): As an example, the leftmost set of boxplots describes the distribution of Physical Health Summary within each bicluster. The variables have been standardized by the procedure described above (page 5), such that 0 denotes the sample mean and the box in each boxplot marks the middle 50% of the values. The interquartile range (IQR) for bicluster #1 (red) stretches from about 0 to 1, indicating that the middle 50% of knees in bicluster #1 have PHS values between 0 and 1 standard deviations above the sample mean. Biclusters #4 (cyan) and #5 (blue) have their middle 50% of knees between 0.5 and 1.5 standard deviations below the mean. Biclusters 3 and 6 do not incorporate the Physical Health Summary variable, so there is no boxplot for those biclusters.

R3Q8: entire manuscript is not well written specifically the methodology & results section; hard to relate the process mentioned in methodology to the results.

Author response: Hopefully the changes made above have addressed the reviewer’s concerns.

R3Q9: Format, mistake in citation style. Please check all. (e.g., “For example, Guan, et al utilized deep learning (DL).. )

Author response/changes made: Citations moved to immediately after et al. in this sentence and the following one, we will confirm all formatting is correct with the editorial team.

---

## [Editor Report · Decision Letter 1]

31 Mar 2022

Biclustering reveals potential knee OA phenotypes in exploratory analyses: Data from the Osteoarthritis Initiative

PONE-D-21-36664R1

Dear Dr. Nelson,

We’re pleased to inform you that your manuscript has been judged scientifically suitable for publication and will be formally accepted for publication once it meets all outstanding technical requirements.

Kind regards,

Shazlin Shaharudin

Academic Editor

PLOS ONE
---

## [Editor Report · Acceptance letter]

13 May 2022

PONE-D-21-36664R1 

Biclustering reveals potential knee OA phenotypes in exploratory analyses: Data from the Osteoarthritis Initiative 

Dear Dr. Nelson:

I'm pleased to inform you that your manuscript has been deemed suitable for publication in PLOS ONE. Congratulations! Your manuscript is now with our production department. 

Kind regards, 

on behalf of

Dr. Shazlin Shaharudin 

Academic Editor

PLOS ONE